

# The influence of sample geometry on the permeability of a porous sandstone

Michael J. Heap[1]

[1]Géophysique Expérimentale, Institut de Physique de Globe de Strasbourg (UMR 7516 CNRS, Université de
Strasbourg/EOST), 5 rue René Descartes, 67084 Strasbourg cedex, France

*Correspondence to*: Michael J. Heap (heap@unistra.fr)

**Abstract.** Although detailed guidelines exist for measuring the physical and mechanical properties of laboratory rock
samples, guidelines for laboratory measurements of permeability are sparse. Provided herein are gas permeability
measurements of cylindrical samples of Darley Dale sandstone (with a connected porosity of 0.135 and a pore- and grain-
10 size of 0.2-0.3 mm) with different diameters (10, 20, and 25 mm) and lengths (from 60 to 10 mm), corresponding to aspect
(length/diameter) ratios between 6.2 and 0.4. These data show that, despite the large range in sample length, aspect ratio, and
bulk volume (from 29.7 to 1.9 cm$^3$), the permeabilities of the Darley Dale sandstone samples are near identical (3-4 × 10$^{-15}$
m$^2$). The near identical permeability of these samples is considered the consequence of the homogeneous porosity structure
typical of porous sandstones, and the small grain- and pore-size of Darley Dale sandstone with respect to the minimum tested
diameter and length (both 10 mm). Laboratory permeability measurements on rock samples with inhomogeneous porosity
structures, or with larger grain- and pore-sizes, may still provide erroneous values if their length, diameter, and/or aspect
ratio is low. Permeability measurements on rocks with vastly different microstructural properties should now be conducted
in a similar manner to help develop detailed guidelines for laboratory measurements of permeability.

## 1 Introduction

Suggested methods and instruments exist for measuring the physical and mechanical properties of rock, such as uniaxial
compressive strength and fracture toughness. For example, the guidelines presented by the International Society for Rock
Mechanics (ISRM; https://www.isrm.net/) and the American Society for Testing and Materials (ASTM;
https://www.astm.org/) are often quoted in experimental papers to assure the community that the measurements were
conducted according to a strict standard. The benefits of such practices are that published experimental data are (1) of a high
standard and (2) can be compared from one publication to another. However, although most laboratory studies of
permeability describe their methods in detail, there is no community consensus on how such measurements ought to be
performed.

Permeability is a measure of the ability of a material to transmit fluids (Guéguen and Palciauskas, 1994). The permeability of
crustal rocks therefore controls the movement of fluids and distribution of pore pressure in the Earth's crust. As a result,



permeability is thought to exert influence over the recurrence of earthquakes (e.g., Sibson, 1992; Caine et al., 1996; Faulkner et al., 2010) and volcanic eruptions (e.g., Eichelberger et al., 1986; Melnik et al., 2005; Mueller et al., 2008; Farquharson et al., 2017; Cassidy et al., 2018), as well as the distribution of ores (e.g., Rowland and Simmons, 2012), the productivity of geothermal reservoirs (e.g., Grant et al., 2013), and the suitability and long-term integrity of $CO_2$ storage sites (e.g.,

Wollenweber et al., 2010).

The permeability of rocks is measured in the laboratory using different methods (steady-state method, transient- or pulse-decay method (e.g., Brace et al., 1968; Mueller et al., 2005), and the oscillating pore pressure method (e.g., Kranz et al., 1990; Fischer and Paterson, 1992)), under different conditions (confining pressure (e.g., Brace et al., 1968; David et al., 1994; Nara et al., 2011) and temperature (e.g., Morrow et al., 2001; Kushnir et al., 2017)), using different pore fluids (liquid

water and gas (e.g., Tanikawa and Shimamoto, 2009; Heap et al., 2018)), and on samples with different geometries (shape, length, and diameter). Different methods and different conditions are used to suit the nature of the rock samples tested and the goal of a particular study. For example, it is impracticable on laboratory timescales to measure low-permeability samples using the steady-state method, and high-pressures are required when studying the permeability structure of the crust. Clearly, guidelines for measuring permeability in the laboratory cannot demand that all values are measured using the same technique

and under the same conditions. However, it is appropriate to form a community consensus as to the factors common to these studies, such as recommendations as to the sample geometry, how a sample is dried (for measurements of gas permeability) or saturated (for measurements of water permeability) prior to measurement, how long a sample should be left at a certain pressure increment before a measurement is taken, which pore fluid should be used, and whether a confining pressure is required to prevent pore fluid from passing between the sample and the sample jacket. The aspect tackled in this study is

sample geometry. For example, when rock samples are rare, small, and/or oddly-shaped, permeability is sometimes measured on samples with geometries that may not be considered entirely appropriate. A minimum aspect ratio of unity is often anecdotally quoted for laboratory measurements of permeability but, to the author's knowledge, no experimental data exist to confirm or deny this "rule of thumb". The goal of this contribution is to better understand, using cylindrical core samples of a widely-used porous sandstone, the influence of sample geometry on laboratory measurements of permeability.

## 25 2 Experimental material

Darley Dale sandstone (Figure 1), a feldspathic sandstone from Derbyshire (England), was chosen for this study due to its wide use in laboratory studies (e.g., Read et al., 1995; Ayling et al., 1995; Zhu and Wong, 1997; Wong et al., 1997; Baud et al., 2000; Wu et al., 2000; Baud et al., 2004; Heap et al., 2009). Darley Dale sandstone has a connected porosity of 0.135 and an average pore- and grain-size of 0.2-0.3 mm (Figure 1). The mineral composition of Darley Dale sandstone (estimated

from a thin section) is 69% quartz, 26% feldspar, 3% clay, and 2% mica (Heap et al., 2009).

Three cylindrical core samples, of diameters 10, 20, and 25 mm, were cored from the same block and in the same direction and were cut and their ends ground-flat and parallel to a nominal length of 60 mm. The three samples were then washed (to

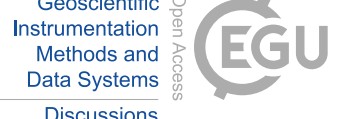



remove any water-soluble grinding fluid) and dried in a vacuum-oven at 40 °C for at least 48 h. The permeabilities of the three samples were then measured as outlined below. Once measured, the length of each of the samples was reduced by 5 mm and the samples were washed, dried, and permeability was re-measured. This process was continued until the samples were 10 mm long (although the 10 mm-diameter sample broke as its length was reduced from 15 to 20 mm). When the 20

5   mm-diameter sample reached a length of 40 mm, five measurements of permeability were performed to ascertain measurement precision. The repeated use of the same three samples, as opposed to > 30 unique samples, was an attempt to avoid permeability variability associated with natural sample heterogeneity.

**3 Method**

Permeability was measured using a benchtop gas (nitrogen) permeameter (Figure 2; Farquharson et al., 2016; Heap and

10   Kennedy, 2016) using the steady-state flow method (selected due to the reasonably high permeability of Darley Dale sandstone). All permeability measurements were conducted under a confining pressure, $P_c$, of 1 MPa and under ambient laboratory temperatures. A confining pressure of 1 MPa was used to prevent gas from passing between the sample and the rubber jacket (Figure 2). Samples were left at 1 MPa for 1 h prior to measurement to ensure microstructural equilibrium.

Volumetric flow rate, $Q_v$, measurements were taken (using a Bronkhorst gas flowmeter with a maximum flow rate of 50

15   ml.min$^{-1}$ and a precision of 0.005 ml.min$^{-1}$) for six different pressure differentials, $\Delta P$ (defined here as the upstream pore fluid pressure, $P_u$, minus the downstream pore fluid pressure, $P_d$). In the permeameter used for this study (Figure 2), $P_d$ is simply the atmospheric pressure (taken here to be 101325 Pa). Values of $\Delta P$ were typically between 0.05 and 0.2 MPa (measured using a pressure transducer with a precision of 5 Pa), equating to flow rates between 5 and 45 ml.min$^{-1}$ (depending on the radius of the sample). Flow rates, and their corresponding pressure differentials, were recorded only when

these values were constant. Assuming laminar flow, the permeability, $k_D$, was then calculated for each $\Delta P$ using the following relation:

$$k_D = \frac{Q_v}{P_m \Delta P} \frac{\mu L P_d}{A},$$   (1)

where $\mu$ is the viscosity of the pore fluid (taken as the viscosity of nitrogen at 20 °C = 1.76 × 10$^{-5}$ Pa·s), $P_m$ is the mean pore fluid pressure (i.e., $(P_u + P_d)/2$), and $A$ and $L$ are the sample cross sectional area and length, respectively. Sample lengths and

diameters were measured using digital callipers (with a precision of 0.005 mm).

The reason for calculating $k_D$ for different values of $\Delta P$ is to assess the Darcian permeability (Equation 1) for fluid flow related artefacts: gas slip along flow channel walls (i.e. the Klinkenberg effect; Klinkenberg, 1941) and/or turbulent flow (i.e. the Forchheimer effect; Forchheimer, 1901). To check whether the Forchheimer correction is required, $1/k_D$ is plotted for each $\Delta P$ as a function of $Q_v$. The Forchheimer correction is deemed necessary if these data are well described by a positive

linear relationship. The Forchheimer-corrected permeability is taken as the inverse of the $y$-intercept of the best-fit linear

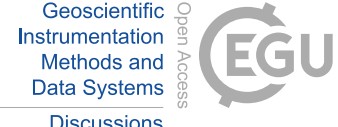



regression of this positive linear relationship. If the Forchheimer correction is required, the data are then checked for the Klinkenberg correction. To do this, $k_{forch}$ is calculated for each $\Delta P$ using:

$$\frac{1}{k_D} = \xi Q_v + \frac{1}{k_{forch}},$$
(2)

where $\xi$, not strictly needed in this analysis, is the slope of the plot of $1/k_D$ as a function of $Q_v$. $k_{forch}$ is then assessed as a

function of $1/P_m$. The Klinkenberg correction is necessary if these data are well described by a positive linear relationship, and the true permeability is taken as the $y$-intercept of the best-fit linear regression of the data. If the data on the plot of $k_{forch}$ as a function of $1/P_m$ cannot be described by a positive linear slope, then the true permeability is taken as the inverse of the $y$-intercept of the best-fit linear regression on the graph of $1/k_D$ as a function of $Q_v$. In the absence of a Forchheimer correction, the need for a Klinkenberg correction is determined by assessing $k_D$ as a function of $1/P_m$. A Klinkenberg correction is

required if these data can be well described by a positive linear relationship. If required, the true sample permeability is given by the $y$-intercept of the best-fit linear regression on the plot of $k_D$ as a function of $1/P_m$. $k_D$ is taken as the true permeability if no corrections are required and is given by the slope of the graph of $Q_v$ as a function of $\Delta P$ multiplied by the mean pore fluid pressure $P_m$. More information on these methods can be found in Heap et al. (2017) and Kushnir et al. (2018).

**4 Results**

Plots of permeability as a function of sample aspect (length/diameter) ratio, bulk sample volume, and sample length are provided in Figures 3a, 3b, and 3c, respectively. These data show that, regardless of sample aspect ratio, volume, and length, the permeability of the measured Darley Dale sandstone samples did not differ significantly from 3-4 $\times$ 10$^{-15}$ m$^2$ (Figure 3; Table 1). The five measurements performed on the sample 20 mm in diameter and 40 mm in length, to ascertain

measurement precision, yielded permeability values of 3.14 $\times$ 10$^{-15}$, 3.17 $\times$ 10$^{-15}$, 3.18 $\times$ 10$^{-15}$, 3.12 $\times$ 10$^{-15}$, and 3.11 $\times$ 10$^{-15}$ m$^2$ (Table 1) (standard deviation of 3.04 $\times$ 10$^{-17}$ m$^2$) and therefore highlight the high precision of the measurements presented.

**5 Discussion, conclusions, and recommendations**

The aim of this study was to better understand, using cylindrical core samples of a widely-used porous sandstone, the

influence of sample geometry on laboratory measurements of permeability. It is often anecdotally considered that an aspect ratio of unity is the minimum for reliable estimates of permeability. Here it is shown that, for Darley Dale sandstone, laboratory measurements of permeability yield the same value over a wide range of sample aspect ratio (from 6.2 and 0.4; Figure 3a), including aspect ratios below unity, bulk volumes (from 29.7 to 1.9 cm$^3$; Figure 3b), and sample lengths (from 60 to 10 mm; Figure 3c). It is likely that this is the result of the small pore- and grain-size (0.2-0.3 mm; Figure 1) with respect to

the minimum tested length/diameter (10 mm) and the homogenous porosity structure of Darley Dale sandstone (Figure 1).



This result is of interest to those studying the permeability of porous sandstones, as it adds confidence to measurements conducted on samples with a small diameter, length, and/or volume. For example, since X-ray computed tomography is often performed on small-diameter cores, permeability modelling using images of these cores could be confidently verified using laboratory measurements on the same cores (e.g., Fredrich et al., 2006). However, samples that contain, for example,

very large pores or inhomogeneously connected porosity structures may still provide erroneous values of permeability if their lengths, diameters, and/or aspect ratios are low. Examples of rocks that are often characterised by complex microstructures include volcanic rocks (e.g., Farquharson et al., 2015). Permeability measurements on rocks with vastly different microstructural properties should now be conducted in a similar manner to help develop detailed guidelines, such as a minimum microstructural feature (grain-size or pore-size) to sample diameter or length ratio, for laboratory measurements

of permeability.

**Data availability**

The data collected for this study are available in Table 1.

**Author contributions**

M.H. conceived the idea for this study, performed the experiments, and wrote the manuscript.

**Competing interests**

The author declares no competing interests.

**Acknowledgements**

Philip Meredith kindly furnished the block of Darley Dale sandstone. Jamie Farquharson and Alex Kushnir are thanked for their role in the development of the benchtop permeameter. Alex Kushnir and Fabian Wadsworth are thanked for
proofreading the manuscript.

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



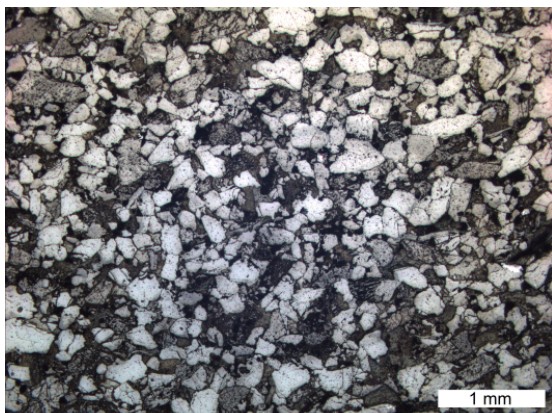

**Figure 1. Microscopic image of Darley Dale sandstone, taken in plane polarised light using an optical microscope.**



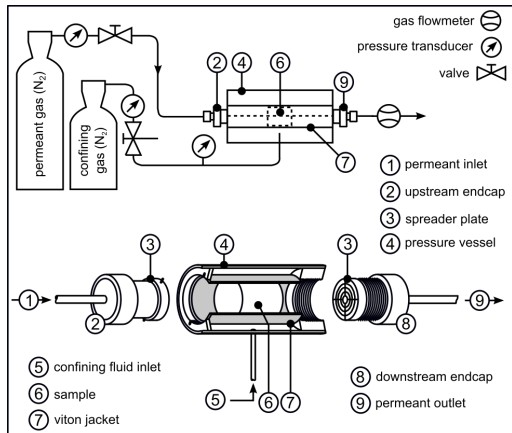

**Figure 2. Schematic of the benchtop gas permeameter used for this study (modified from Farquharson et al., 2016; Heap and Kennedy, 2016).**





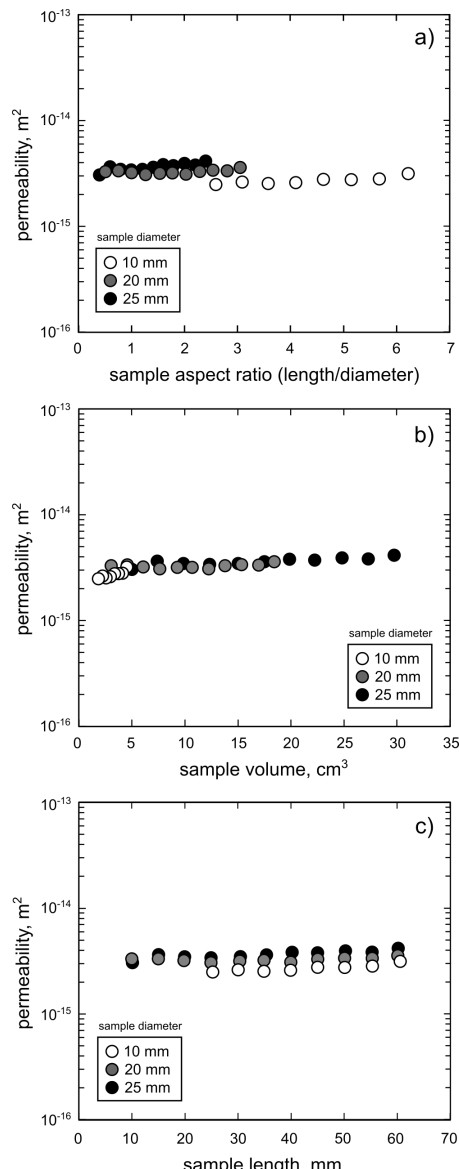

**Figure 3. Permeability of Darley Dale sandstone as a function of sample aspect (length/diameter) ratio (a), bulk sample volume (b), and sample length (c). Errors associated with transducer precision are encapsulated by the symbol size.**





**Table 1. Summary of the experimental data collected for this study.**

| Length (mm) | Diameter (mm) | Aspect ratio (length/diameter) | Bulk volume (cm$^3$) | Permeability (m$^2$) | Correction used |
|---|---|---|---|---|---|
| 60.54 | 9.74 | 6.22 | 4.51 | $3.16 \times 10^{-15}$ | none |
| 55.32 | 9.74 | 5.68 | 4.12 | $2.83 \times 10^{-15}$ | none |
| 50.10 | 9.74 | 5.14 | 3.73 | $2.78 \times 10^{-15}$ | none |
| 44.98 | 9.74 | 4.62 | 3.35 | $2.78 \times 10^{-15}$ | none |
| 39.93 | 9.74 | 4.10 | 2.98 | $2.61 \times 10^{-15}$ | none |
| 34.85 | 9.74 | 3.58 | 2.60 | $2.54 \times 10^{-15}$ | Forchheimer |
| 30.03 | 9.74 | 3.08 | 2.24 | $2.63 \times 10^{-15}$ | Forchheimer |
| 25.23 | 9.74 | 2.59 | 1.88 | $2.50 \times 10^{-15}$ | Forchheimer |
| 60.12 | 19.74 | 3.05 | 18.40 | $3.60 \times 10^{-15}$ | Forchheimer |
| 55.29 | 19.74 | 2.80 | 16.92 | $3.37 \times 10^{-15}$ | Forchheimer |
| 50.06 | 19.74 | 2.54 | 15.32 | $3.38 \times 10^{-15}$ | Forchheimer |
| 45.01 | 19.74 | 2.28 | 13.78 | $3.31 \times 10^{-15}$ | Forchheimer |
| 39.98 | 19.74 | 2.03 | 12.24 | $3.14 \times 10^{-15}$ | Forchheimer |
| 39.98 | 19.74 | 2.03 | 12.24 | $3.17 \times 10^{-15}$ | Forchheimer |
| 39.98 | 19.74 | 2.03 | 12.24 | $3.18 \times 10^{-15}$ | Forchheimer |
| 39.98 | 19.74 | 2.03 | 12.24 | $3.12 \times 10^{-15}$ | Forchheimer |
| 39.98 | 19.74 | 2.03 | 12.24 | $3.11 \times 10^{-15}$ | Forchheimer |
| 34.89 | 19.74 | 1.77 | 10.68 | $3.22 \times 10^{-15}$ | Forchheimer |
| 30.33 | 19.74 | 1.54 | 9.28 | $3.18 \times 10^{-15}$ | Forchheimer |
| 24.91 | 19.74 | 1.26 | 7.62 | $3.09 \times 10^{-15}$ | Forchheimer |
| 19.79 | 19.74 | 1.00 | 6.06 | $3.23 \times 10^{-15}$ | Forchheimer |
| 14.98 | 19.74 | 0.76 | 4.58 | $3.36 \times 10^{-15}$ | Forchheimer |
| 9.98 | 19.74 | 0.51 | 3.05 | $3.31 \times 10^{-15}$ | Forchheimer |
| 60.17 | 25.08 | 2.40 | 29.73 | $4.18 \times 10^{-15}$ | Forchheimer |
| 55.24 | 25.08 | 2.20 | 27.29 | $3.84 \times 10^{-15}$ | Forchheimer |
| 50.23 | 25.08 | 2.00 | 24.81 | $3.96 \times 10^{-15}$ | Forchheimer |
| 44.99 | 25.08 | 1.79 | 22.23 | $3.78 \times 10^{-15}$ | Forchheimer |
| 40.19 | 25.08 | 1.60 | 19.85 | $3.85 \times 10^{-15}$ | Forchheimer |
| 35.37 | 25.08 | 1.41 | 17.47 | $3.62 \times 10^{-15}$ | Forchheimer |
| 30.43 | 25.08 | 1.21 | 15.03 | $3.48 \times 10^{-15}$ | Forchheimer |
| 24.93 | 25.08 | 0.99 | 12.32 | $3.41 \times 10^{-15}$ | Forchheimer |
| 19.89 | 25.08 | 0.79 | 9.83 | $3.48 \times 10^{-15}$ | Forchheimer |
| 14.98 | 25.08 | 0.60 | 7.40 | $3.66 \times 10^{-15}$ | Forchheimer |
| 10.10 | 25.08 | 0.40 | 4.99 | $3.08 \times 10^{-15}$ | Forchheimer |