# Peer review of "The influence of sample geometry on the permeability of a porous sandstone"

_Geoscientific Instrumentation, Methods and Data Systems, 2018_

## Referee Comment (RC1) · M. Colombier (Referee) · 2 Nov 2018

The author of "The influence of sample geometry on the permeability of a porous sandstone" studies the permeability fluctuations for rock samples of a similar origin (sandstone) for different lengths, diameters and aspect ratios. The author shows that, under the range of sample volume and aspect ratio taken into consideration, there are very limited variations of permeability. This study therefore provides results of broad interest for researchers studying permeability of rocks and is an essential step towards a community consensus for permeability measurements.

The manuscript is clearly written, easy to follow. To my opinion, the manuscript is already suitable as it isfor publication. I only provide a couple of minor comments for

the author.

Minor comments

Last page of the discussion, line 2-4 The author discuss the use of X-ray computed tomography on small-diameter cores and proposes that laboratory measurements of permeability might bring confidence on the results of XCT. Studies of pore structure using XCT frequently focus on even smaller rock samples (diameter and height <5mm). Would the author expect that such volumes would still be representative for the Darley Dale sandstones studied here? Or does the author expect a scattering of permeability values at volumes smaller than those analyzed in this study? At which critical volume would this scattering occur? It already looks from Figure 3b that there is slightly more variability of the permeability at low volume of interest

Line 5 Pore size, shape, aperture size, anisotropy, tortuosity and connectivity will likely all have a role for other types of samples.
* * *

---

## Short Comment (SC1) · 4 Dec 2018

I have reviewed the submitted paper titled 'The influence of sample geometry on the permeability of a porous sandstone' that reports systematic permeability data measured in samples of a homogeneous, fine grain sandstone as a function of the cylinder's aspect ratio.

The article reads very well. However, one has to be cautious regarding the suggested larger impact of the study compared to its actual content. If the data presented are well supporting the discussion and concluding remarks written in section 5, the abstract and introduction invite the readers to expect a much larger demonstration.

The method and data presented are for benchmarking the usage of a benchtop gas

permeameter although there is a need, as well pointed by the author, for a standard description on how to perform high quality permeability test in laboratory in the best reproducible manner. Indeed, accessing to permeability value in triaxial test rigs for instance can be done using different sample geometry (cylinders, cubes), of various sizes and aspect ratios, and with different type of fluids (liquid, gas) and methods (flow, pulse-decay, oscillations).

I would therefore recommend to be a bit more precise in stating clearly that this paper deals with rock matrix permeability first, and also on benchtop measurements of this parameter. The sample geometry mentioned could also be simplify to aspect ratio for immediate clarity.

On a personal note, as I fully support the broader scope wished with that study, I would recommend the author to discuss with Prof. Christian David (Uni. Cergy-Pontoise, France) and Prof. Patrick Selvadurai (McGill University, Canada), who both trialled large permeability measurement benchmarking few years ago. It is my hope that the results they may have gathered could help the author to pursue this study.

Few other points: -Page 3, line 5: why only 1 sample once was tested 5 times and not other? -Page 3, line 13: "for 1 h prior to measurement to ensure microstructural equilibrium" How does the author know/control that this time was sufficient for the mentioned purpose? -The author presents both the Forchheimer and the Klinkenberg corrections. In Table 1, one can see that the Forchheimer correction has been applied to most of the measurements. Yet it is not stated clearly anywhere why the Klinkenberg correction was not needed. An additional figure demonstrating for 1 test at least why the Forchheimer correction was needed and how it was calculated would be of great value as well.

---

## Author Comment (AC1) · 18 Dec 2018

Reviewer #1 (Mathieu Colombier)

The author of "The influence of sample geometry on the permeability of a porous sandstone" studies the permeability fluctuations for rock samples of a similar origin (sandstone) for different lengths, diameters and aspect ratios. The author shows that, under the range of sample volume and aspect ratio taken into consideration, there are very limited variations of permeability. This study therefore provides results of broad interest for researchers studying permeability of rocks and is an essential step towards a community consensus for permeability measurements.

The manuscript is clearly written, easy to follow. To my opinion, the manuscript is

already suitable as it is for publication. I only provide a couple of minor comments for the author.

>We thank the reviewer for these endorsements.

Minor comments

Last page of the discussion, line 2-4 The author discuss the use of X-ray computed tomography on small-diameter cores and proposes that laboratory measurements of permeability might bring confidence on the results of XCT. Studies of pore structure using XCT frequently focus on even smaller rock samples (diameter and height <5mm). Would the author expect that such volumes would still be representative for the Darley Dale sandstones studied here? Or does the author expect a scattering of permeability values at volumes smaller than those analyzed in this study? At which critical volume would this scattering occur? It already looks from Figure 3b that there is slightly more variability of the permeability at low volume of interest.

>It is questions like these that I'd hoped to stimulate from the permeability-measuring community. In the absence of experimental data, my guess would be that the permeability of Darley Dale sandstone cores with lengths and radii shorter than the cores measured herein would be the same. Given that the pore size of Darley Dale sandstone is on the order of 200 microns, and that the pore structure is very homogenous, cores 2-3 mm in length and/or 2-3 mm in diameter may be sufficient (i.e. volumes between 0.006 and 0.02 cm3). I've now added an additional sentence to the discussion section to this effect:

>"Based on the small pore size and homogenous pore structure of Darley Dale sandstone, core samples smaller than those measured herein (e.g. samples with diameters and/or lengths of 2 or 3 mm) may be sufficient to obtain reliable values of permeability."

>I must stress, however, that this prediction is for Darley Dale sandstone only: rocks with a larger pore size would surely require larger samples (an interesting hypothesis

to test) and rocks with the same pore size as Darley Dale sandstone, but a higher porosity, may also require larger samples (another interesting hypothesis to test). Experimentally, there's plenty of room for further study.

>Regarding Figure 3b, I agree that it appears as though the permeability of the 10 mm-diameter sample is decreasing as the sample volume is decreased. However, this is largely a result of the semi-log scale: the permeabilities of the 10 mm-diameter samples are, when one considers the expected variability between different measurements, essentially identical (see the listed values in Table 1).

Line 5 Pore size, shape, aperture size, anisotropy, tortuosity and connectivity will likely all have a role for other types of samples.

>Agreed. This is discussed towards the end of the discussion section. I've now added an additional reference that deals with the complexity of pore connectivity in volcanic rocks (Colombier et al., 2017):

>"However, samples that contain, for example, very large pores or inhomogeneously connected porosity structures may still provide erroneous values of permeability if their lengths, diameters, and/or aspect ratios are low. Examples of rocks that are often characterised by complex microstructures include volcanic rocks (e.g., Farquharson et al., 2015; Colombier et al., 2017)."

---

## Author Comment (AC2) · 18 Dec 2018

Reviewer #2 Audrey Ougier-Simonin

I have reviewed the submitted paper titled 'The influence of sample geometry on the permeability of a porous sandstone' that reports systematic permeability data measured in samples of a homogeneous, fine grain sandstone as a function of the cylinder's aspect ratio. The article reads very well. However, one has to be cautious regarding the suggested larger impact of the study compared to its actual content. If the data presented are well supporting the discussion and concluding remarks written in section 5, the abstract and introduction invite the readers to expect a much larger demonstration. The method and data presented are for benchmarking the usage of a

benchtop gas permeameter although there is a need, as well pointed by the author, for a standard description on how to perform high quality permeability test in laboratory in the best reproducible manner. Indeed, accessing to permeability value in triaxial test rigs for instance can be done using different sample geometry (cylinders, cubes), of various sizes and aspect ratios, and with different type of fluids (liquid, gas) and methods (flow, pulse-decay, oscillations). I would therefore recommend to be a bit more precise in stating clearly that this paper deals with rock matrix permeability first, and also on benchtop measurements of this parameter. The sample geometry mentioned could also be simplify to aspect ratio for immediate clarity.

>One of the goals of the introduction was to highlight, for the benefit of a general readership, why permeability warrants our close attention. A short paragraph in the introduction highlights that permeability is thought to influence earthquake and volcanic eruption recurrence, the distribution of ores, the productivity of geothermal resources, and the suitability of $CO_2$ storage sites. Based on the title of the paper ("The influence of sample geometry on the permeability of a porous sandstone"), and the discussion that follows this short paragraph, I'm confident that those reading the paper will not jump to the conclusion that I'm trying to solve any of the aforementioned geophysical phenomena. Indeed, I state the goal of the paper very clearly at the end of the introduction:

>"The goal of this contribution is to better understand, using cylindrical core samples of a widely-used porous sandstone, the influence of sample geometry on laboratory measurements of permeability."

>I use "sample geometry", rather than "aspect ratio", because the manuscript also shows the influence of sample length and sample volume on laboratory measurements of permeability, not just sample aspect ratio.

On a personal note, as I fully support the broader scope wished with that study, I would recommend the author to discuss with Prof. Christian David (Uni. Cergy-Pontoise,

[Figure]

France) and Prof. Patrick Selvadurai (McGill University, Canada), who both trialled large permeability measurement benchmarking few years ago. It is my hope that the results they may have gathered could help the author to pursue this study.

>I'm pleased by the reviewer's interest in the study. I'm aware of the as-of-yet-unpublished studies of Christian David and Patrick Selvadurai. In fact, Thierry Reuschlé, a member of the laboratory at Strasbourg to which I am affiliated, is involved in both studies. I eagerly await the results of these studies.

Few other points:

Page 3, line 5: why only 1 sample once was tested 5 times and not other?

>The permeability of each sample was measured at least twice. It is the average of these measurements that is presented in the manuscript. This has now been clarified in the manuscript:

>"Once measured (each sample was measured at least twice; an average of these values is presented herein), the length of each of the samples was reduced by 5 mm and the samples were washed, dried, and permeability was re-measured."

>One sample, 20 mm in diameter and 40 mm in length, was measured five times, and the measurements shown in the manuscript, to inform the reader as to the precision of the measurements presented. This is explained in the manuscript:

>"When the 20 mm-diameter sample reached a length of 40 mm, five measurements of permeability were performed to ascertain measurement precision."

Page 3, line 13: "for 1 h prior to measurement to ensure microstructural equilibrium" How does the author know/control that this time was sufficient for the mentioned purpose?

>This is a very good question. The pore volume (porosity) of a water-saturated sample in our triaxial setup, measured using a pore fluid pressure intensifier, typically takes

about 30 minutes or so to stabilise under a given confining pressure. It is assumed that the microstructure has equilibrated when the porosity reaches a plateau. However, this timescale cannot be verified/measured using our benchtop gas permeameter. To be on the safe side, I decided to leave the samples at 1 MPa for 1 hour prior to measurement. As stated in the introduction, there is no community consensus as to the time required for sample equilibration prior to a measurement of permeability, offering another avenue for future research. The time I left the samples at the target pressure prior to measurement is provided in the submitted manuscript in the interests of transparency. However, I've now reworded this sentence to avoid the word "ensure":

>"Samples were left at 1 MPa for 1 h prior to measurement to allow for microstructural equilibrium."

The author presents both the Forchheimer and the Klinkenberg corrections. In Table 1, one can see that the Forchheimer correction has been applied to most of the measurements. Yet it is not stated clearly anywhere why the Klinkenberg correction was not needed. An additional figure demonstrating for 1 test at least why the Forchheimer correction was needed and how it was calculated would be of great value as well.

>This is a good point. I've now added a new figure that shows why the Forchheimer correction was needed, and why the Klinkenberg correction was not (now Figure 3; see "Fig. 1" below). This figure has been woven into the text to help the reader better understand how I tested whether these corrections were needed:

>"The Forchheimer correction is deemed necessary if these data are well described by a positive linear relationship (an example is shown in Figure 3a; these data highlight that a Forchheimer correction is needed). The Forchheimer-corrected permeability is taken as the inverse of the y-intercept of the best-fit linear regression of this positive linear relationship."

>"If the data on the plot of kforch as a function of 1/Pm cannot be described by a positive linear slope, as in the example shown in Figure 3b, then the true permeability

is taken as the inverse of the y-intercept of the best-fit linear regression on the graph of 1/kD as a function of Qv (i.e. the best-fit linear regression shown in Figure 3a)."

>"For the data collected for this study, either no correction or the Forchheimer correction was needed (see Table 1). The Klinkenberg correction was not required for any of the measurements (Table 1). More information on these methods can be found in Heap et al. (2017) and Kushnir et al. (2018)."

>Caption for the revised Figure 3 ("Fig. 1" below):

>"Figure 3. (a) The reciprocal of Darcian permeability, kD, as a function of volumetric flow rate (for the sample 25 mm in diameter and 60 mm in length). The data can be well described by a positive linear slope: the Forchheimer correction is therefore needed. (b) The Forchheimer-corrected permeability as a function of the reciprocal of the mean pore fluid pressure (for the sample 25 mm in diameter and 60 mm in length; the same experiment shown in panel (a)). Since these data cannot be well-described by a positive linear slope, no Klinkenberg correction is required."
* * *
[Figure]

**Fig. 1.**